# Towards Cross-Sample Alignment for Multi-Modal Representation Learning in Spatial Transcriptomics

**Justina Dai** [*†]
jusdai@ethz.ch

**Kalin Nonchev**[*†]
kalin.nonchev@inf.ethz.ch

**Viktor H Koelzer** [‡§¶]
viktor.koelzer@usb.ch

**Gunnar Rätsch**[†¶]
raetsch@inf.ethz.ch

## Abstract

The growing number of spatial transcriptomics (ST) datasets enables comprehensive multi-modal characterization of cell types across diverse biological and clinical contexts. However, integration across patient cohorts remains challenging, as local microenvironment, patient-specific variability, and technical batch effects can dominate signals. Here, we hypothesize that combining specialized transcriptomics correction methods with deep representation learning can jointly align morphology, transcriptomics, and spatial information across multiple tissue samples. This approach benefits from recent transcriptomics and pathology foundation models, projecting cells into a shared embedding space where they cluster by cell type rather than dataset-specific conditions. Applying this framework to 18 skin melanoma, 12 human brain, and 4 lung cancer datasets, we demonstrate that it outperforms conventional batch-correction approaches by 58%, 38%, and 2-fold, respectively. Together, this framework enables efficient integration of multi-modal ST data across modalities and samples, facilitating the systematic discovery of conserved cellular programs and spatial niches while remaining robust to cohort-specific batch effects.

**Code availability**: https://github.com/ratschlab/aestetik

## 1 Introduction

Recent advances in spatial transcriptomics (ST) enable the simultaneous mapping of tissue morphology and gene expression while preserving spatial context Ståhl et al. (2016); Chelebian et al. (2025). Although individual, donor-specific ST datasets provide incremental insights into cellular organization and local microenvironments, constructing comprehensive ST atlases requires integrating data both *vertically* across modalities and *horizontally* across patients.

Integrating cells *vertically* across molecular profiles and morphology features is important for gaining a multi-modal perspective on cellular interactions in normal physiology and disease Rao et al.

---

[*]Equal contribution.

[†]Institute for Machine Learning, Department of Computer Science, ETH Zurich, Switzerland

[‡]Institute of Medical Genetics and Pathology Group, University Hospital of Basel, Basel, Switzerland

[§]Computational and Translational Pathology Group, Department of Biomedical Engineering, University of Basel, Basel, Switzerland

[¶]Equal supervision.

(2021); Yoosuf et al. (2020). Recent deep-learning approaches have demonstrated that combining transcriptomics, morphology, and spatial information enhances the identification of spatially defined biological niches and their associated pathways, leading to a more refined understanding of tissue organization and function Nonchev et al. (2024); Bao et al. (2022); Xu et al. (2022). However, these methods are typically applied on a per-sample basis (e.g., 6.5 mm$^2$ tissue capture area in 10x Visium), fragmenting analyses and limiting the detection of biological patterns conserved across patients Hu et al. (2024).

Meanwhile, ST datasets are rapidly expanding Jaume et al. (2024); Chen et al. (2024a); Xu et al. (2024); Consortium & Hoffmann (2025), making the *horizontal* integration of cells across samples, patients, and studies critical for uncovering general spatial niches, conserved biomarkers, and cellular programs across clinical conditions. Although existing transcriptomics batch-correction algorithms have been successful for gene expression data Lopez et al. (2018); Hie et al. (2019); Korsunsky et al. (2019), their adaptation to spatially resolved transcriptomics data remains under-explored. This limitation restricts the correction of batch effects across molecular and morphology modalities while preserving spatial context.

To this end, we hypothesize that leveraging the strengths of *vertical* multi-modal deep representation learning together with the effectiveness of *horizontal* cross-sample alignment methods will enhance the identification of biologically meaningful spatial niches across ST atlases. Specifically, applying this framework to embeddings derived from recent foundation models in transcriptomics and pathology further improves their representation, enabling the construction of comprehensive multi-modal ST atlases spanning multiple donors.

We evaluate this hypothesis on a diverse ST benchmark consisting of 12 human brain samples, 18 melanoma samples, and 4 lung cancer samples. To assess both within-patient and cross-patient alignment, we employ two complementary evaluation strategies: single-donor integration, which evaluates adjacent tissue sections from the same donor, and multi-donor integration, which evaluates samples across different donors. The results demonstrate consistently more accurate, coherent, and biologically meaningful domain assignments than any batch-correction method alone. Ablation studies further show that these gains arise from spatial multi-modal integration, with the model effectively leveraging multiple data modalities rather than relying on any single source.

Beyond spatial domain identification, these vertically and horizontally integrated representations support downstream analyses of both morphology and gene expression. Notably, these findings highlight the emerging need for flexible, multi-modal computational frameworks capable of integrating diverse ST datasets, providing a foundation for future efforts to map cellular organization and tissue-level interactions across biological scales.

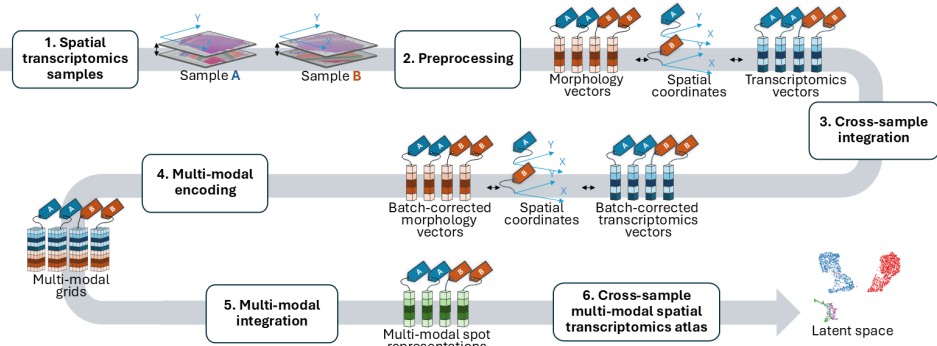

Figure 1: Graphical abstract. Multiple ST samples are preprocessed and horizontally integrated via cross-modal batch correction. The data are then structured into multi-modal grids for vertical integration using deep representation learning, yielding a multi-modal ST atlas.

## 2 RELATED WORK

### 2.1 CELL REPRESENTATIONS

In transcriptomics, cells are represented as high-dimensional gene expression vectors, which are often reduced to low-dimensional embeddings to reveal cellular heterogeneity and biological structure. Principal Component Analysis (PCA) is commonly used for this purpose due to its simplicity and efficiency, but it may miss complex, non-linear relationships Heumos et al. (2023). To address some of these limitations, recent transcriptomics foundation models have been developed to generate richer non-linear embeddings that capture subtle transcriptomics patterns Cui et al. (2024a;b); Hao et al. (2024). Building on this idea, specialized oncology models such as CancerFoundation Theus et al. (2024), trained exclusively on malignant cells, produce embeddings tailored to cancer datasets, facilitating improved performance in downstream analyses.

### 2.2 SPATIAL TRANSCRIPTOMICS BATCH EFFECTS

The rapidly growing collection of ST datasets Jaume et al. (2024); Chen et al. (2024a); Xu et al. (2024); Consortium & Hoffmann (2025) is improving our understanding of cellular heterogeneity and tissue-level organization. However, these datasets are high-dimensional and sparse, and they often contain technical artifacts, making the extraction of meaningful biological patterns challenging. These issues are particularly pronounced in ST, where small capture areas (e.g., 6.5 mm$^2$ in 10x Genomics) typically contain tissue from a single patient. Consequently, observed signals may reflect local microenvironments, technical variation, or patient-specific factors such as genetics or disease state, limiting interpretability and generalization across samples.

### 2.3 TRANSCRIPTOMICS CORRECTION METHODS ACROSS SAMPLES

Unsupervised transcriptomics batch-correction methods are designed to *horizontally* integrate data from multiple conditions while mitigating batch effects, without requiring prior knowledge of cell-type labels Heumos et al. (2023). These methods aim to both harmonize datasets and separate technical artifacts from genuine biological variability. Notable examples include:

- scVI implements a probabilistic framework using a hierarchical variational autoencoder, modeling gene expression as a negative binomial distribution and explicitly incorporating batch covariates into the latent space to separate batch-driven variation from biological signal Lopez et al. (2018).

- Scanorama applies a manifold alignment strategy inspired by panorama stitching in computer vision: it identifies mutual nearest neighbors across datasets, computes linear correction vectors in gene expression space, and iteratively merges datasets while preserving local structure Hie et al. (2019).

- Harmony operates on low-dimensional embeddings (e.g., PCA), iteratively performing soft clustering and adjusting cell embeddings to maximize batch mixing within clusters, thereby aligning datasets while maintaining biologically meaningful differences Korsunsky et al. (2019).

Despite their effectiveness in correcting batch effects in transcriptomics, these methods are inherently non-spatial. They do not incorporate physical tissue context or spatial relationships between cells, limiting their applicability to spatial transcriptomics data, where location-dependent effects are critical.

### 2.4 REPRESENTATION LEARNING ACROSS ST MODALITIES

Recent advances in *vertical* representation learning for ST aim to jointly capture gene expression, spatial context, and, in some cases, morphology features to uncover meaningful multi-modal cellular and tissue-level patterns. A variety of models have been proposed, such as MUSE, which uses a multi-view autoencoder to learn a shared latent space from transcriptomics and morphology features Bao et al. (2022). BayesSpace uses a Bayesian framework that places greater prior weight on

spatially neighboring spots Zhao et al. (2021). Alternative methods use graph neural networks to construct a unified three-dimensional neighborhood graph Li et al. (2024).

The *recent* AESTETIK method Nonchev et al. (2024) introduces two key advances. First, it jointly models morphology, transcriptomics, and spatial information by constructing spatial grids that can be adaptively initialized. Second, its efficient convolutional autoencoder architecture, combined with a self-supervised multi-sample triplet loss, makes it scalable to millions of cells. Importantly, the flexibility of AESTETIK's loss function allows assessment of each modality's contribution, as well as the relative importance of spatial information.

## 2.5 MORPHOLOGY REPRESENTATIONS

In image-based ST, each spot corresponds to a small tissue region (e.g., 55 μm of tissue in 10x Visium). Traditionally, such regions have been encoded using general-purpose image foundation models trained on natural images Bao et al. (2022); Nonchev et al. (2024). While these models can extract general visual features, they are not tailored to histopathology and may fail to capture fine-grained tissue structures relevant for biological interpretation. This limitation motivates the use of pathology foundation models (PFMs), which are trained on large-scale histopathology datasets using self-supervised techniques such as contrastive learning or masked image modeling Chen et al. (2024b); Saillard et al. (2024).

## 3 METHODS

We developed a general framework for integrating multi-modal ST data, combining gene expression, tissue morphology, and spatial context to robustly identify conserved spatial niches and cellular programs across multiple samples and donors.

## 3.1 MULTI-MODAL INTEGRATION OF SPATIAL TRANSCRIPTOMICS DATA ACROSS SAMPLES

Each ST spot $i$ is represented by a transcriptomics vector $\mathbf{x}_i \in \mathbb{R}^{n_t}$, a morphology vector $\mathbf{m}_i \in \mathbb{R}^{n_m}$, and spatial coordinates $\mathbf{s}_i = (x_i, y_i)$. To mitigate technical and donor-specific variation, batch correction is applied independently using established methods, including Harmony Korsunsky et al. (2019), scVI Lopez et al. (2018), or Scanorama Hie et al. (2019). The procedure incorporates the sample or donor identity as a covariate to separate technical variation from biological signals. For transcriptomics, we evaluated multiple methods:

$$\tilde{\mathbf{x}}_i = \text{BatchCorrect}(\mathbf{x}_i \mid \text{sample}_i), \tag{1}$$

while for morphology, batch correction was applied exclusively using Harmony:

$$\tilde{\mathbf{m}}_i = \text{BatchCorrect}(\mathbf{m}_i \mid \text{sample}_i), \tag{2}$$

producing corrected features $\tilde{\mathbf{x}}_i$ and $\tilde{\mathbf{m}}_i$.

The corrected features are integrated into a unified multi-modal representation using the AESTETIK framework Nonchev et al. (2024). First, the principal components of transcriptomics and morphology features are concatenated, and augmented with local spatial neighborhoods to form a tensor image-like grid

$$\text{spot}_i \in \mathbb{R}^{N_{\text{grid}} \times N_{\text{grid}} \times 2n_{\text{PCA}}}. \tag{3}$$

Spot embeddings are then computed as

$$\mathbf{z}_i = f_{\text{AESTETIK}}(\tilde{\mathbf{x}}_i, \tilde{\mathbf{m}}_i, \mathbf{s}_i), \tag{4}$$

where $f_{\text{AESTETIK}}(\cdot)$ is a convolutional autoencoder trained with a composite loss:

$$\mathcal{L}_{\text{AESTETIK}} = \alpha \cdot \left( \mathcal{L}_{\text{MSE}}^{\text{m}} + \mathcal{L}_{\text{triplet}}^{\text{m}} \right) + (3 - \alpha) \cdot \left( \mathcal{L}_{\text{MSE}}^{\text{tr}} + \mathcal{L}_{\text{triplet}}^{\text{tr}} \right), \tag{5}$$

with $\alpha \in [0, 3]$ controlling the relative contributions of morphology ('m') and transcriptomics ('tr'). The multi-triplet loss encourages spots with similar labels to be close in embedding space while pushing dissimilar spots apart, producing robust multi-modal representations. Positive and negative examples for the loss are defined based on precomputed modality-specific clusters using K-Means, enabling self-supervised learning without requiring ground-truth labels.

## 3.2 SPATIAL DOMAIN IDENTIFICATION

The learned embeddings $\{\mathbf{z}_i\}_{i=1}^{N}$ are clustered to define tissue domains:

$$\mathcal{C} = \text{Cluster}(\{\mathbf{z}_i\}_{i=1}^{N}),$$

with K-Means as the default method. Cluster assignments can be refined using K-nearest neighbor majority voting on spatial coordinates to ensure spatial continuity.

**Code availability**   AESTETIK is implemented in Python 3 and is available as open source software at http://www.github.com/ratschlab/aestetik. The Snakemake pipeline for reproducing the results is available at https://github.com/ratschlab/multi-slide-aestetik-2026-analysis.

## 3.3 EVALUATION AND BENCHMARKING OF MULTI-MODAL ST ACROSS SAMPLES

To avoid hyperparameter tuning on test data, we used a variant of nested cross-validation (nCV) tailored to the hierarchical structure of our data, where spots are nested within samples and samples within donors. To prevent data leakage, samples were first grouped by donor, forming donor-level partitions.

For each outer nCV fold, donor groups were split into a tuning set and a test set. Hyperparameters were optimized on the tuning set via a two-step grid search: (1) candidate parameters were evaluated by learning spot-level representations across all tuning samples, performing joint clustering, and computing a composite performance score; (2) the best-performing parameters were then evaluated on the held-out test set.

We benchmark multi-modal integration using two complementary tasks:

**Single-donor integration**   This task evaluates the framework's ability to integrate samples from the same donor, such as adjacent tissue slices. The tuning set consisted of a single donor group. For testing, each donor group was processed independently: spot representations were learned jointly using the selected hyperparameters, followed by clustering and evaluation within the same donor group. The LIBD human dorsolateral prefrontal cortex dataset was used for this task (Appendix A.2).

**Multi-donor integration**   This task evaluates the integration of samples across different donors. The tuning set included multiple donor groups. For testing, all samples were processed jointly: spot representations were learned across the entire test set with the selected hyperparameters, followed by joint clustering and evaluation across donors. The Tumor Profiler melanoma and USZ lung cancer datasets were used for this task (Appendix A.2).

**Integration quality metrics**   Integration performance during hyperparameter optimization is quantified using a composite score suggested in Luecken et al. (2021) that balances biological conservation and batch mixing:

$$\text{Score} = \lambda \cdot \text{mean}(\mathcal{M}_{\text{bio}}) + (1 - \lambda) \cdot \text{mean}(\mathcal{M}_{\text{batch}}),$$

where $\mathcal{M}_{\text{bio}}$ includes metrics such as ARI, NMI, and silhouette score, $\mathcal{M}_{\text{batch}}$ includes iLISI and kBET, and $\lambda \in [0, 1]$ controls the relative weighting.

## 4 EXPERIMENTS

### 4.1 INTEGRATING MULTI-MODAL DATA ACROSS PATIENTS IMPROVES SPATIAL DOMAIN IDENTIFICATION

Figure 2 demonstrates the impact of incorporating spatial multi-modal embeddings in cross-sample integration. We compared Harmony, Scanorama, and scVI applied to transcriptomics data alone versus their versions augmented with spatial multi-modal representations, using datasets with ground-truth tissue annotations. Performance was measured by the Adjusted Rand Index (ARI), with the number of clusters set to the true number of labels to ensure comparable conditions across models.

Across all datasets and under both integration settings, joint cross-sample integration augmented with spatial multi-modal embeddings consistently produced clusters that were substantially closer to the biological ground truth compared to analyses performed using transcriptomics data alone. For example, in the human brain dataset (top), integrating adjacent samples with multi-modal data from the same donor increased the mean ARI by 32% for Harmony, 64% for Scanorama, and 20% for scVI (Figure 10). A similar pattern was observed for multi-donor integrations. The strongest effect was seen in lung cancer (bottom left), where joint integration resulted in a two-fold increase in mean ARI compared with analyzing samples independently after transcriptomics batch correction, increasing ARI from 0.18 in scVI to 0.5 in scVI+AESTETIK. Notably, deriving morphology representations with specialized pathology foundation models (e.g., UNI2-h) improved domain identification relative to general-purpose image models (e.g., Inception v3) (Figure 12).

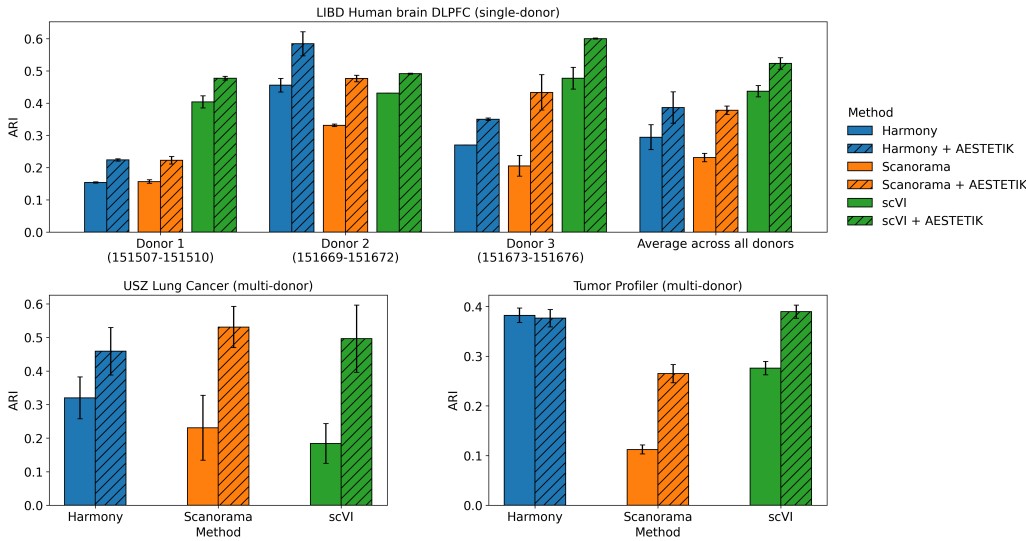

Figure 2: Benchmark results comparing the performance of the proposed cross-sample multi-modal approach with cross-sample transcriptomics integration alone. The y-axis represents the ARI between the ground truth and the predicted labels.

Figure 3 qualitatively illustrates the latent-space integration for non-integrated data, transcriptomics-only integration (Harmony), and multi-modal integration (Harmony+AESTETIK) across two lung cancer patients. In non-integrated data, spots cluster mainly by patient rather than biological identity. With Harmony, donor-specific clustering is reduced but still noticeable. In Harmony+AESTETIK, which integrates morphology, transcriptomics, and spatial information, the multi-triplet loss brings similar embeddings together and pushes dissimilar ones apart, resulting in correct clustering of normal tissue (top left), tumor tissue (top right), and tertiary lymphoid structures (bottom left), with nearby inflamed spots. Cells from both donors cluster well by domain, although some patient-specific effects remain, likely reflecting inter-patient variability.

Furthermore, a pathway analysis of the Harmony+AESTETIK clusters was performed using decoupleR Badia-i Mompel et al. (2022) (Figure 13), which revealed increased activity of the PI3K and MAPK pathway in the tumor cluster, consistent with their roles in promoting cell growth, proliferation, and survival Dinsmore & Soriano (2018). Additionally, increased WNT activity was observed in the bottom left cluster, known for repair and regeneration in lung development Aros et al. (2021).

Overall, these results demonstrate that leveraging spatially informed multi-modal representations for cross-sample integration substantially improves the biological coherence and accuracy of tissue domain identification relative to non-spatial transcriptomics batch-correction methods alone.

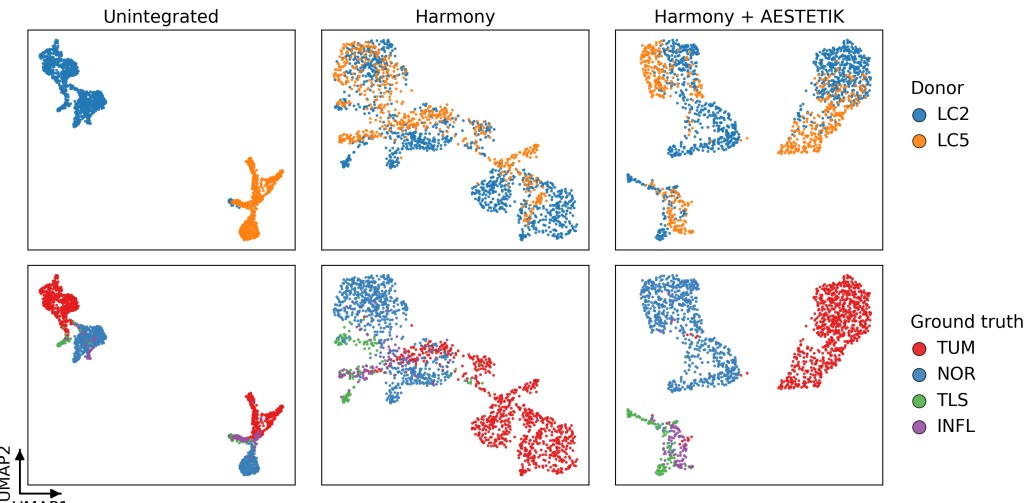

Figure 3: UMAP comparison of ST data integration across non-integrated, transcriptomics-only (Harmony) and multi-modal integration across samples (Harmony+AESTETIK) in lung cancer.

## 4.2 REPRESENTATIONS DERIVED FROM TRANSCRIPTOMICS FOUNDATION MODELS IMPROVE ST INTEGRATION

Next, we investigated whether replacing the linear transcriptomics PCA projection used in batch correction methods with recent transcriptomics foundation models (e.g., CancerFoundation) improves performance in spatial domain identification, and whether incorporating morphology and spatial information provides additional gains. Figure 4 compares results across different configurations (standalone, with Harmony, and with Harmony plus AESTETIK). We observe that both PCA-only and CancerFoundation-only embeddings exhibit pronounced donor-specific effects in skin melanoma and lung cancer samples. Applying Harmony to these embeddings reduces donor effects and increases the biological signal, improving the alignment of shared tissue structures (e.g., CancerFoundation in Tumor Profiler - ARI (ground truth) $0.06 \rightarrow 0.22$). Furthermore, including spatial and morphology information boosts the biological signal even more, with CancerFoundation outperforming PCA across both datasets when combined with multi-modal cross-sample integration.

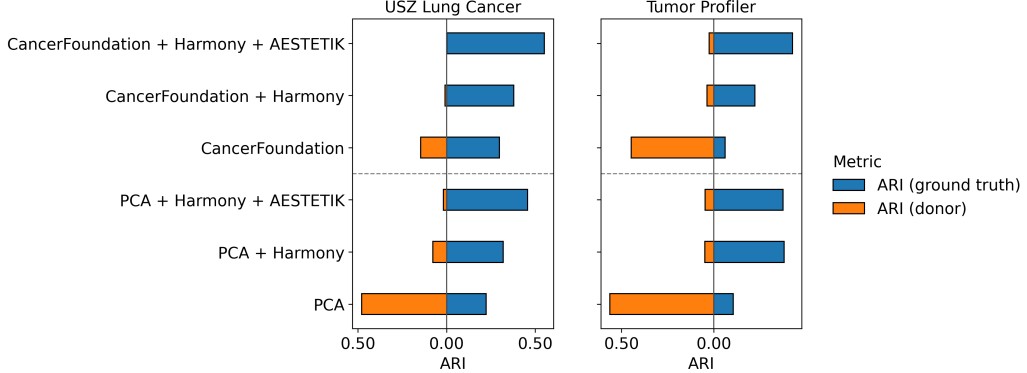

Figure 4: Benchmarking cross-sample multi-modal integration using transcriptomics foundation models. (1) Higher ARI ground truth indicates better cluster consistency with ground truth, (2) lower ARI donor reflects improved donor mixing.

## 4.3 ABLATION STUDY

Lastly, to highlight the importance of spatial information, we systematically varied the grid size in AESTETIK on the LIBD Human DLPFC dataset, ranging from 1 (no spatial context) to 7, and measured the resulting change in ARI (Figure 5). The grid size determines how many spatially adjacent spots are considered. Across all batch-correction models, local spatial information was beneficial for preserving fine-grained details and spot-to-spot variability. Performance peaked at a window size of 5, while increasing it to 7 offered no improvement, suggesting that overly large spatial contexts dilute local signals.

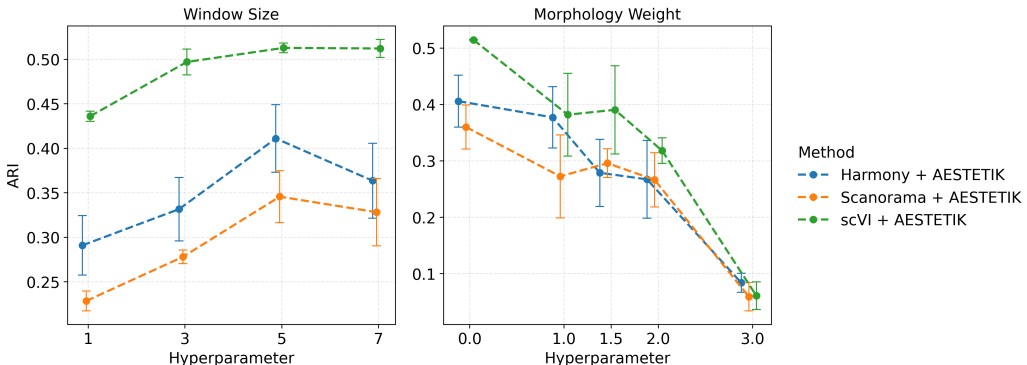

Figure 5: Ablation study of window size and morphology weight effects on ARI in the brain dataset.

Similarly to Nonchev et al. (2024), we confirmed that the transcriptomics data is informative in the brain dataset, as it closely reflects the known cytoarchitecture and marker gene expression consistent with the ground-truth annotations Maynard et al. (2021) (Figure 5).

## 5 DISCUSSION & CONCLUSION

Our results demonstrate that combining *horizontal* transcriptomics batch correction with *vertical* spatially informed multi-modal representation learning substantially improves the integration of ST data across both adjacent single-donor and multi-donor samples. Unlike conventional batch-correction approaches that primarily target transcriptomics, our framework incorporates morphology and spatial context, enabling the alignment of shared tissue structures while preserving key biological and donor-specific signals.

Across diverse datasets, including human brain, melanoma, and lung cancer, spatial multi-modal embeddings improved the alignment of shared tissue structures, as reflected by higher ARI scores and more coherent latent-space clustering (Figures 2, 3). These results highlight that local spatial context and tissue morphology provide complementary information to transcriptomics, particularly in tissues with complex architectures, such as layered neurons or heterogeneous tumors.

Notably, our framework can leverage recent advances in transcriptomics and pathology foundation models (e.g., CancerFoundation, UNI) to enhance the detection of biological signals and spatial domains, complementing existing representations. Furthermore, we show that preserving local spatial neighborhoods is essential for maintaining spatial biological niches across samples.

Overall, this framework offers a scalable and flexible platform for constructing multi-modal ST atlases, facilitating the discovery of conserved cellular programs and spatial niches across patients and conditions.

## 6 LIMITATION & FUTURE WORK

Although our framework shows strong potential for improving the integration of multi-modal ST data, there remain important opportunities for further refinement. One major challenge is separating technical artifacts from true biological differences and patient-specific effects, especially in tumors.

More work is needed to make sure that important biological signals are not lost during integration. In our study, we integrated morphology images in latent space using Harmony to align them with transcriptomics, but future work should systematically compare this approach to traditional stain normalization methods, such as Macenko Macenko et al. (2009) or Reinhard Magee et al. (2009), to assess robustness across tissues and imaging protocols.

While our evaluation focused on datasets with available expert annotations, our results already demonstrate the method's robust performance. Future studies can further extend this work by applying the framework to emerging single-cell and high-resolution spatial transcriptomics technologies, such as Visium HD, to explore its capabilities across diverse datasets and gain deeper insights into its generalizability and potential applications.

Looking ahead, it may be beneficial to develop a single machine learning framework that combines batch correction with multi-modal representation learning, rather than handling them separately. This could allow the model to learn more complex tissue patterns in an end-to-end way. Additionally, more in-depth analysis of downstream tasks, such as identifying spatial domains and conserved cellular programs, will help show the full value of multi-modal embeddings and guide improvements to the method.

## 7 MEANINGFULNESS STATEMENT

This work learns meaningful representations of biological systems by integrating molecular measurements with tissue morphology and spatial organization, which are central to biological function. We address a challenge in spatial transcriptomics where tissue samples are often limited to single specimens, causing clustering by patient-specific variability and batch effects instead of cell type. Our framework aligns shared tissue structures across samples while preserving biologically relevant spatial neighborhoods, producing representations that capture conserved niches rather than technical artifacts. By leveraging spatial context and multi-modal information from unimodal biological models, these representations offer a richer view of biological processes and disease dynamics.

**Acknowledgments** Computational data analysis was performed at Leonhard Med (https://sis.id.ethz.ch/services/sensitiveresearchdata/), a secure trusted research environment at ETH Zurich.

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

## A APPENDIX

### A.1 HYPERPARAMETERS

For an individual model, the hyperparameter search space is defined as the set of all possible combinations of its parameters listed in Table 1. When multiple models are combined, the hyperparameter search space is defined as the Cartesian product of each individual model's parameter sets.

| Model | Hyperparameters |
|---|---|
| scVI | n_layers $\in \{1, 2, 3, 4\}$ |
| Harmony | theta $\in \{0, 0.5, 1, 2, 4, 6, 8\}$ |
| Scanorama | sigma $\in \{10, 15, 20, 25\}$ |
| AESTETIK | morphology_weight : $[0, 1, 1.5, 2, 3]$
window_size : $[5, 7]$
refine_cluster : {True, False} |
| CancerFoundation | — |

Table 1: Model hyperparameters

Default hyperparameters were used unless otherwise specified.

### A.2 DATASETS

**Human brain LIBD dataset**  The LIBD dataset Maynard et al. (2021), sequenced with 10x Visium, consists of 12 tissue slices from the dorsolateral prefrontal cortex (DLPFC) region. The slices were obtained from three donors, with two pairs of spatially adjacent replicates. Each slice was manually annotated into six or four distinct neuronal layers and white matter (WM) by considering cytoarchitecture and selected gene markers.

**Tumor Profiler dataset**  The Tumor Profiler dataset Nonchev et al. (2024) contains 18 tissue slices of metastatic melanoma sequenced using the 10x Visium platform. The slices originate from 9 distinct tissue regions ($6.5 \times 6.5$ mm$^2$) across 7 donors, with 2 replicates per region. Each spot was first classified using histopathology software into one of the categories tumor, stroma, normal lymphoid, blood/necrosis and subsequently reviewed by a pathologist.

**USZ Lung cancer dataset**  The USZ Lung cancer dataset Nonchev et al. (2025) comprises 5 tissue slices of lung cancer with tertiary lymphoid structures (TLS) sequenced using the 10x Visium platform. Each tissue slice was obtained from a different patient. Expert researchers manually annotated spots as TLS, Immune, Tumor, Normal, Lymph nodes, with remaining spots categorized as Unassigned. One sample (LC4) was excluded from analysis due to low quality.

| Donor | Region | Sample ID |
|-------|--------|-----------|
| Donor 1 | Anterior | 151507 151508 |
| | Posterior | 151509 151510 |
| Donor 2 | Anterior | 151669 151670 |
| | Posterior | 151671 151672 |
| Donor 3 | Anterior | 151673 151674 |
| | Posterior | 151675 151676 |

Table 2: LIBD Human DLPFC dataset hierarchy

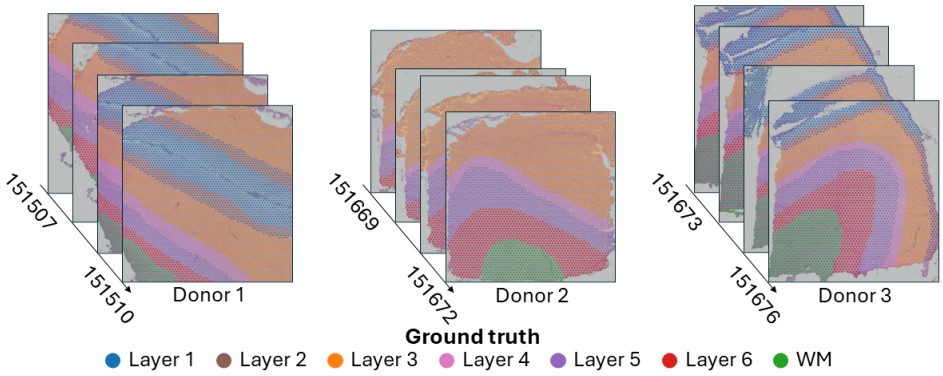

Figure 6: LIBD Human DLPFC dataset with manual ground truth annotations

## A.3 DATASET HIERARCHY

## A.4 DATA AVAILABILITY

The USZ lung cancer dataset was downloaded from Zenodo https://zenodo.org/records/14620362, the LIBD brain dataset https://research.libd.org/spatialLIBD/, and the Tumor Profiler spatial transcriptomics dataset upon request at Nonchev et al. (2024). The image foundation model weights for UNI2-h are available at HuggingFace https://huggingface.co/MahmoodLab/UNI2-h and Inception v3 on PyTorch.

## A.5 DATA PREPROCESSING

Raw gene expression matrices were first concatenated across all samples. Genes expressed in fewer than 10 spots were then filtered out. Highly variable genes were subsequently identified by selecting the top 3,000 genes using ScanpyWolf et al. (2018) with the Seurat v3 flavor, using the sample identifier as a batch key. For batch correction with scVI, the resulting gene expression was directly used as input. Otherwise, the data was additionally normalized to target sum 10,000 (1e4), log-transformed, and scaled. Finally, Principal Component Analysis (PCA) was applied to the counts, extracting the top 20 PCs.

Transcriptomics preprocessing for embeddings derived with CancerFoundation follows the procedures described in the tutorial available in the project's public GitHub repository.

| Donor | Region | Sample ID |
|-------|--------|-----------|
| MACEGEJ | Region 1 | MACEGEJ-1-1
MACEGEJ-1-2 |
| | Region 2 | MACEGEJ-2-1
MACEGEJ-2-2 |
| MAHEFOG | Region 1 | MAHEFOG-1-1
MAHEFOG-1-2 |
| MAJOFIJ | Region 1 | MAJOFIJ-1-1
MAJOFIJ-1-2 |
| | Region 2 | MAJOFIJ-2-1
MAJOFIJ-2-2 |
| MAKYGIW | Region 1 | MAKYGIW-1-1
MAKYGIW-1-2 |
| MANOFYB | Region 1 | MANOFYB-1-1
MANOFYB-1-2 |
| MELIPIT | Region 1 | MELIPIT-1-1
MELIPIT-1-2 |
| MIDEKOG | Region 1 | MIDEKOG-1-1
MIDEKOG-1-2 |

Table 3: Tumor Profiler dataset hierarchy

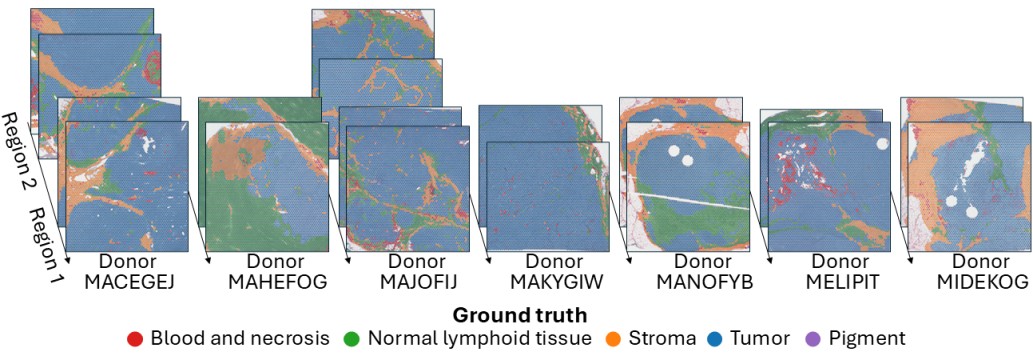

Figure 7: Tumor Profiler dataset with ground truth annotations

### A.5.1 PATHOLOGY EMBEDDINGS

The steps for extracting morphology embeddings are illustrated in Figure 9.

First, spot-centered image patches were extracted from each tissue slice, with the patch size defined by the spot diameter. Features were then extracted using two models:

- Inception v3 was used with pretrained ImageNet weights. Patches were first resized to 299, then center-cropped, and subsequently normalized following the standard Inception v3 pre-processing. Features were extracted from the penultimate layer (with 2,048 dimensions).

- UNI2-h with pretrained weights from the project's HuggingFace repository was used as a comparative foundation model. Following the default preprocessing steps of UNI2-h, the patches were first resized to 224 and then normalized using ImageNet channel-wise mean and standard deviation. Features were extracted from the embedding layer (1,536 dimensions).

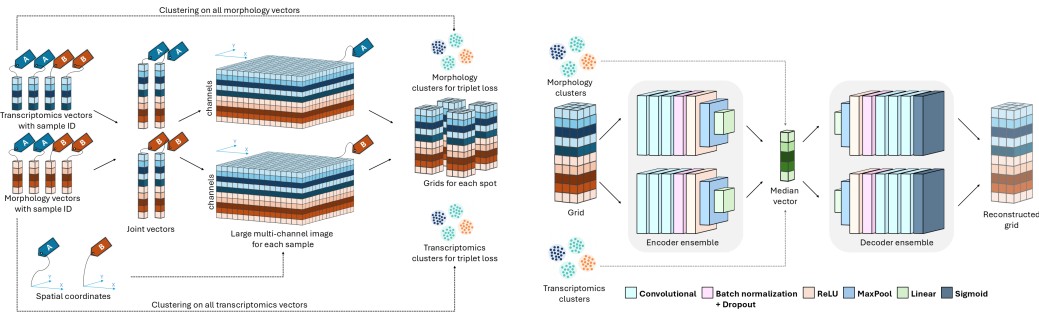

(a) Cross-sample data preprocessing       (b) Model architecture of AESTETIK

Figure 8: Overview of the cross-sample multi-modal integration framework

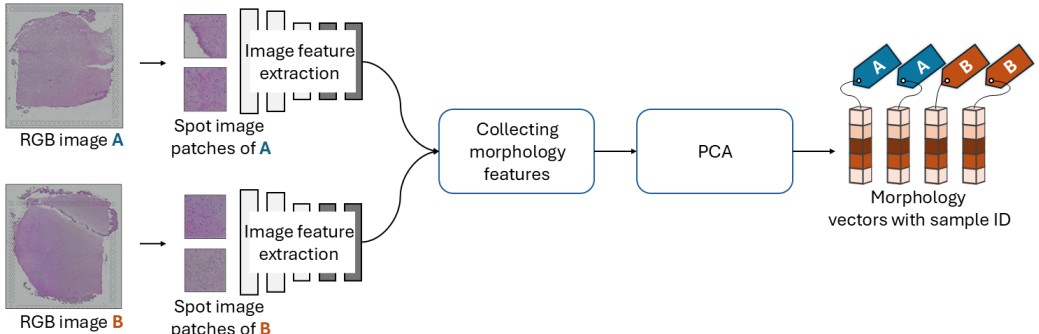

Figure 9: Pathology embedding workflow. (1) Spot patches were cropped from multiple tissue images, (2) features were extracted using pretrained models, (3) collected together, and (4) reduced with PCA.

After feature extraction, the morphology features were aggregated across samples, and principal component analysis (PCA) was used to reduce the feature dimensionality to 20. Performance comparison is available in Figure 12.

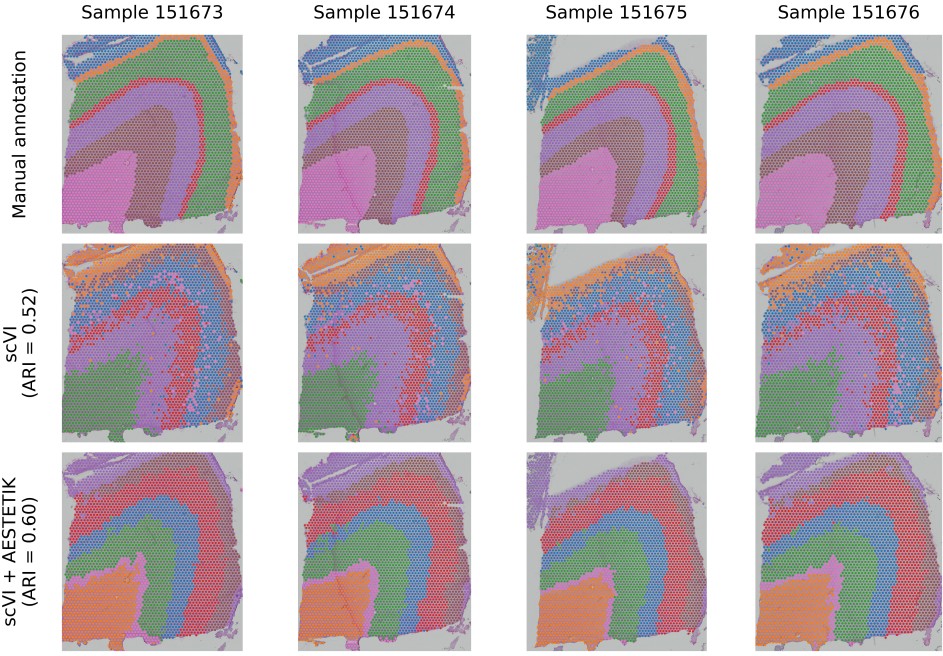

Figure 10: Qualitative comparison on identifying spatial layers in the brain.

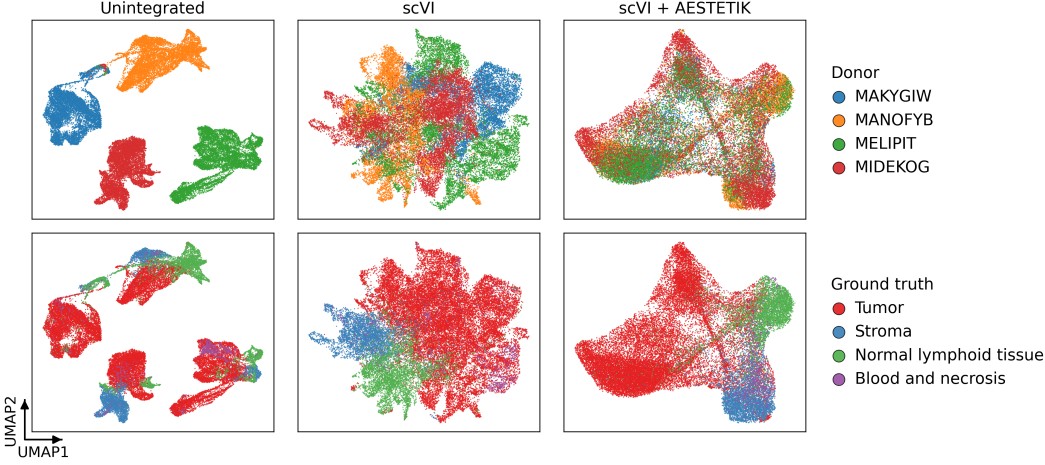

Figure 11: UMAP comparison of ST data integration across non-integrated, transcriptomics-only (scVI) and multi-modal integration (scVI+AESTETIK) across Tumor Profiler samples.

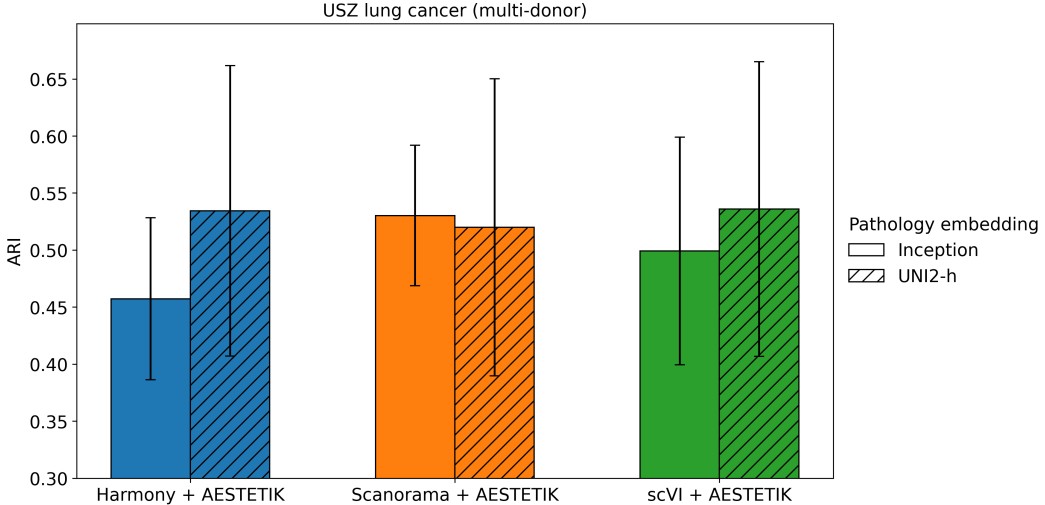

Figure 12: Benchmarking Harmony+AESTETIK, Scanorama+AESTETIK, and scVI+AESTETIK with morphology features extracted using Inception v3 versus UNI2-h on the lung cancer dataset.

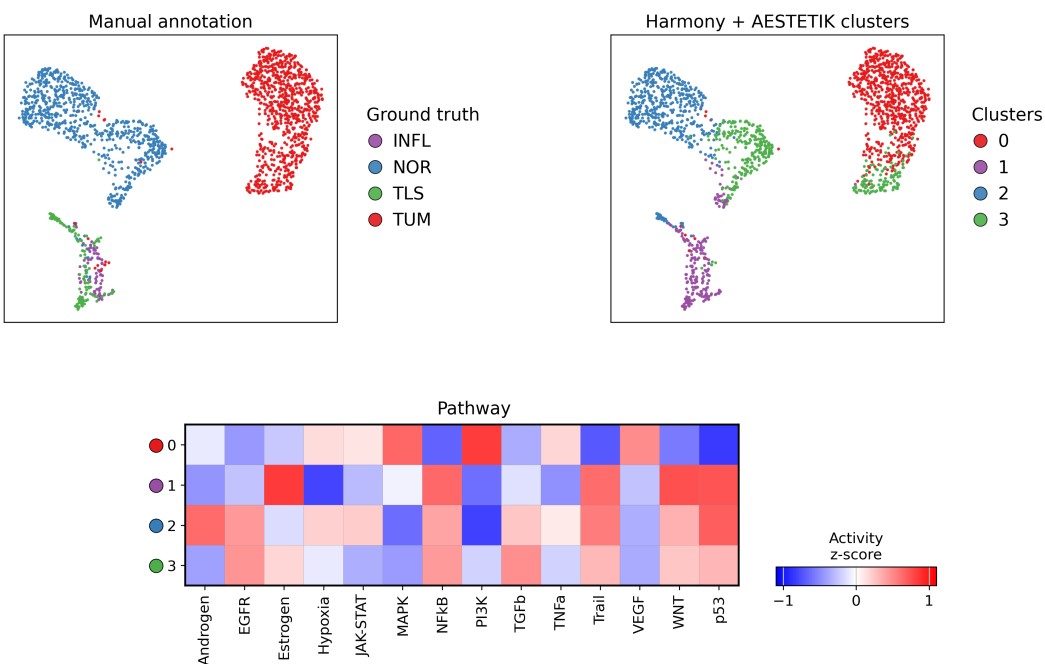

Figure 13: UMAP of multi-modal integration (Harmony+AESTETIK) colored by ground truth (top left) and cluster (top right), with pathway analysis of the identified clusters using decoupleR Badia-i Mompel et al. (2022) in lung cancer.

