# OpenReview forum: "Towards Cross-Sample Alignment for Multi-Modal Representation Learning in Spatial Transcriptomics"
_ICLR.cc/2026/Workshop/LMRL — ICLR 2026 Workshop LMRL Poster_

### Official Review · Reviewer_vAiP · 2026-02-24
**Review of "Towards Cross-Sample Alignment for Multi-Modal Representation Learning in Spatial Transcriptomics"**

**Rating:** 5
**Confidence:** 3

**Review:**

Here the authors propose a pipeline for integrating spatial transcriptomics (ST) datasets. The pipeline is based on combining the outputs obtained from popular methods for scRNA-seq batch effect correction (e.g. Harmony [1]) with a previously proposed representation learning method for spatial omics data (AESTETIK [2]) . The authors find that this combination leads to improved performance on batch effect correction metrics for ST data compared to using the outputs of the scRNA-seq integration methods alone.

These results may represent initial preliminary work; however, the topic of data integration for ST has been extensively studied by many other works which are not cited and/or compared against by the others. For example, spaVAE [3], GraphST [4], and STAGATE [5] among others have capabilities for performing spatially-aware data integration and should be considered in any benchmarks for ST integraiton methods. Moreover, given that the method proposed in this submission is a relatively straightforward combination of existing works, the novelty here is relatively limited.

For these reasons I believe this paper is borderline (if space allows) or a rejection.

References:

[1]: Korsunsky, Ilya, et al. "Fast, sensitive and accurate integration of single-cell data with Harmony." Nature methods 16.12 (2019): 1289-1296.

[2]: Nonchev, Kalin, et al. "Representation learning for multi-modal spatially resolved transcriptomics data." medRxiv (2024): 2024-06.

[3]: Tian, Tian, et al. "Dependency-aware deep generative models for multitasking analysis of spatial omics data." Nature Methods 21.8 (2024): 1501-1513.

[4]: Long, Yahui, et al. "Spatially informed clustering, integration, and deconvolution of spatial transcriptomics with GraphST." Nature communications 14.1 (2023): 1155.

[5]: Dong, Kangning, and Shihua Zhang. "Deciphering spatial domains from spatially resolved transcriptomics with an adaptive graph attention auto-encoder." Nature communications 13.1 (2022): 1739.

---

### Official Review · Reviewer_2Y2R · 2026-02-24
**Interesting approach but extremely preliminary result and missing key information**

**Rating:** 4
**Confidence:** 4

**Review:**

This work combines with-in modality batch removal methods with multi-model integration to improve cross-sample alignment. I think the approach is interesting, but I feel the results are too preliminary to draw any meaningful conclusion. Furthermore, I think the writing can be significantly improved, there are some key information that seems to be missing.

Pros:
- interesting approach to combine with-in modality batch-removal with multi-modal embeddings

Cons:
- Missing explanation of ARI score: for Figure 2, what kind of ground truth labels are the ARI computed for? I don't think this was mentioned in the main text. This is a major issue. Without this information, it's impossible to asses whether this result is biologically meaningful.
- for Figure 3: I'm not sure why the Harmony + AESTETIK task is necessarily better than Harmony alone? the normal spots are usually tissue samples adjacent to tumor, do we really expect the spots to be highly distinct?
- Figure 4: it's not clear that the CancerFoundation model is that much better compared to PCA, especially when AESTETIK is involved.
- Figure 5: if I'm reading this correctly, the right figure seems to suggest that morphology is actually harmful to integration? as opposed to what is claimed in the paper. I'm confused why the authors claim that morphology is useful.
- Finally, why define a Integration quality metrics, but only use ARI in all the figures?

---

### Official Review · Reviewer_Sr8v · 2026-02-25
**Review for Towards Cross-Sample Alignment for Multi-Modal Representation Learning in Spatial Transcriptomics**

**Rating:** 7
**Confidence:** 5

**Review:**

The authors develop a general framework for integrating spatial transcriptomic modalities: expression, morphology and spatial location. The authors validate the hypothesis that combining “horizontal” transcriptomic batch correction (via methods like Harmony) with “vertical” multi modal representation learning improves understanding of spatial transcriptomics data, across three diverse datasets.

**Pros:**

The paper is well written, thorough, and it defines and approaches an important task. Overall, the method seems to largely succeed at that task.

**Cons:**

 The benchmarks presented are in general very basic, for example, differentiating a few brain layers, or 4 cell types in a tumor, so the actual performance or benefits here are not really fully understood.

scFMs are supposed to generally work in a zero shot setting, it is very unusual to apply an integration method like harmony on top of their representations.

How much over integration is happening? Surely there are some differences between donors that the models should include, but in figure 4, for example, the final Donor ARI drops to almost 0.

---

### Meta-Review · Area_Chair_RDZK · 2026-02-25

**Recommendation:** Accept (Poster)
**Confidence:** 4

**Metareview:**

Accept

---

### Decision · Program_Chairs · 2026-03-02

**Decision:**

Accept (Poster)

**Comment:**

Please see the meta-review.